



# Extensive palaeo-surfaces beneath the Evans-Rutford region of the West Antarctic Ice Sheet control modern and past ice flow.

Charlotte M. Carter[1,2], Michael J. Bentley[1], Stewart S. R. Jamieson[1], Guy J. G. Paxman[1], Tom A. Jordan[3], Julien A. Bodart[3,4], Neil Ross[5], Felipe Napoleoni[4]

[1]Durham University, Department of Geography, Lower Mountjoy, South Road, Durham, DH1 3LE, United Kingdom
[2]Alfred-Wegener-Institute, Am Alten Hafen 26, 27568 Bremerhaven, Germany
[3]British Antarctic Survey, High Cross, Madingley Road, Cambridge, CB3 0DT, United Kingdom
[4]University of Edinburgh, School of Geosciences, Drummond Street, Edinburgh, EH8 9XP, United Kingdom
[5]Newcastle University, School of Geography, Politics and Sociology, Claremont Road, Newcastle Upon Tyne, NE1 7RU, United Kingdom

*Correspondence to*: Charlotte Carter (charlotte.carter@awi.de)

**Abstract.**

The subglacial landscape of Antarctica records and influences the behaviour of its overlying ice sheet. However, in many places, the evolution of the landscape and its control on ice sheet behaviour has not been investigated in detail. Using recently released radio-echo sounding data, we investigate the subglacial landscape of the Evans-Rutford region of West Antarctica. Following quantitative analysis of the landscape morphology under ice-loaded and unloaded conditions, we identify ten flat surfaces distributed across the region. Across these ten surfaces, we identify two distinct populations based on clustering of elevations, which potentially represent remnants of regionally coherent pre-glacial surfaces underlying the West Antarctic Ice Sheet (WAIS). The surfaces are bounded by deeply incised glacial troughs, some of which have potential tectonic controls. We assess two hypotheses for the evolution of the regional landscape: (1) passive margin evolution associated with the breakup of the Gondwana supercontinent, or (2) an extensive planation surface that may have been uplifted either in association with the West Antarctic Rift System or cessation of subduction at the base of the Antarctic Peninsula. We suggest that passive margin evolution is most likely of these two mechanisms, with the erosion of glacial troughs adjacent to, and incising, the flat surfaces likely having coincided with the growth of the WAIS. These flat surfaces also demonstrate similarities to other identified surfaces, indicating that a similar formational process may have been acting more widely around the Weddell Sea Embayment. The subsequent fluctuations of ice flow, basal thermal regime and erosion patterns of the WAIS are therefore controlled by the regional tectonic structures.

## 1 Introduction

Subglacial topography is an important boundary condition influencing ice sheet behaviour across Antarctica. The subglacial landscape is not static in time, but evolves via erosion, deposition and tectonic processes. As the landscape changes, ice flow patterns can change in response. Some patterns of landscape promote selective linear erosion, whereby incision in deep troughs





occurs directly adjacent to elevated areas where little or no erosion occurs (Sugden, 1978). Such patterns can therefore be reinforced, as incision deepens valleys over time. However, the initiation of selective glacial incision at the onset of Antarctic glaciation is heavily preconditioned by the existing subaerial topography, which in itself is a function of processes such as

35     tectonics, fluvial erosion and deposition, and hillslope processes (Jamieson and Sugden, 2008). Therefore, if we wish to understand the past and present ice-flow conditions of Antarctica, it is important to explore the subglacial landscape and the processes that have led to its development.

We aim to understand the evolution of the subglacial topography beneath the Evans-Rutford region of the West Antarctic Ice Sheet (WAIS). This region (Figure 1) hosts significant ice streams draining the WAIS and the Antarctic Peninsula, and contains

40     some of the highest relief topography in Antarctica. However, its landscape evolution, as recorded in the detailed subglacial geomorphology, has not been fully explored.

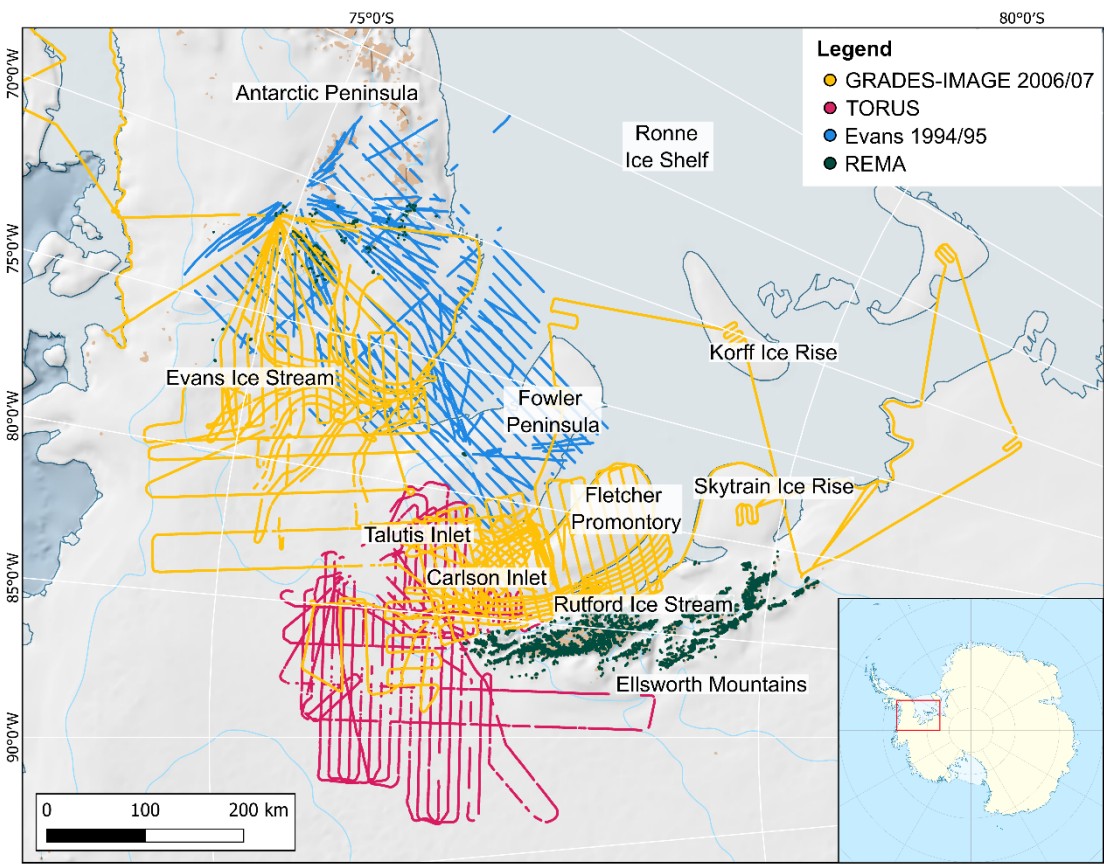

**Figure 1**: Study site map of the Evans-Rutford region. Data points illustrate the RES surveys used in the study: GRADES-IMAGE 2006/07 carried out by British Antarctic Survey (BAS), TORUS, REMA exposed bedrock, and Evans 1994/95.

45     Detailed basemap sourced from Quantarctica (Matsuoka et al., 2021). Inset map shows location of the study region in relation to the Antarctic continent (red box).



The Evans-Rutford region is located between the base of the Antarctic Peninsula and the Ellsworth Mountains (Figure 1). This area is drained by two major outlet glaciers, the Evans and Rutford ice streams, which discharge into the Ronne Ice Shelf. Carlson Inlet contains a stagnated ice stream adjacent to Rutford Ice Stream (King, 2011), whilst the Fletcher Promontory and Fowler Peninsula are slow-flowing ice-rises (Matsuoka et al., 2015), between the ice streams.

Ice flows from the high ice surface topography of the upper catchment of the Evans Ice Stream and from the Ellsworth Mountains (Figure 2a; 2c). The Evans and Rutford ice streams occupy troughs up to 2200 m deep (Figures 2b; 2c). The region is bounded by two areas of alpine-like topography, including over deepened U-shaped valleys and headwall cirques: at the base of the Antarctic Peninsula to the northeast and the Ellsworth Mountains to the southwest. The two ice streams have significantly faster ice flow (> 600 m/yr) close to the grounding line, in contrast with slow-flowing and almost stagnant ice overlying the higher elevation points (> approx. 400m elevation) and within Carlson Inlet (Figure 2c).



**Figure 2**: Glaciology and topography of the study area (horizontal raster mesh resolution of grids stated in brackets). (a) RAMP2 ice surface elevation model (200 m) (Liu et al., 2015), (b) BedMachine bed elevation model (500 m) (Morlighem et al., 2020), (c) MEaSUREs ice flow velocity (450 m) (Rignot et al., 2017), (d) BedMachine ice thickness (1 km) (Morlighem et al., 2020).

Although there have been local-scale glaciological investigations into the physical properties, dynamics and subglacial environments of the Rutford Ice Stream (Smith et al. 2015, King et al., 2016, Minchew et al. 2018, Schlegel et al., 2021), studies of the subglacial topography of the wider region are limited or pre-date modern survey data (Doake et al., 1983). Therefore, to understand the regional subglacial topography, we used the GRADES-IMAGE radio-echo sounding (RES) survey conducted by the British Antarctic Survey (BAS) during the 2006/07 season, in combination with the BAS 1994/95 Evans and 2001/02 TORUS (Targeting ice stream Onset Regions and Under-ice Systems) RES surveys. These campaigns provided approximately 48,000 line-km of coverage over parts of the Antarctic Peninsula, Ellsworth Mountains, Evans and Rutford ice streams, as well as opportunistic surveys of the Korff and Skytrain ice rises (Corr, 2021; Frémand et al., 2023). Airborne RES data compilations have been used to produce gridded bed elevation models for the Weddell Sea sector of the WAIS, including in Bedmap2 and BedMachine (Fretwell et al., 2013; Morlighem et al., 2020; Jeofry et al., 2018), but the detail of the regional landscape has not been interpreted in the context of landscape evolution. Previous utilisation of the RES data from the GRADES-IMAGE survey have led to the identification of 'sticky spots' beneath the Evans Ice Stream (Ashmore et al., 2014), as well as water piracy between Rutford Ice Stream and Carlson Inlet from the TORUS data (Vaughan et al., 2008).

## 1.1 Tectonic history of the Evans-Rutford region

Geologically, the study area lies at the junction of the three main provinces forming West Antarctica (Figure 3); Antarctic Peninsula-Thurston Island, the Weddell Sea Province, and the West Antarctic Rift System-Marie Byrd Land (Jordan et al., 2020). The Weddell Sea Province comprises the marine Weddell Sea Rift System, together with the onshore highlands of the Haag Nunataks block and the Ellsworth-Whitmore Mountains block. The Weddell Sea Province was one of the regions of Antarctica most impacted by the break-up of Gondwana initiated in the Jurassic (Jordan et al., 2020).



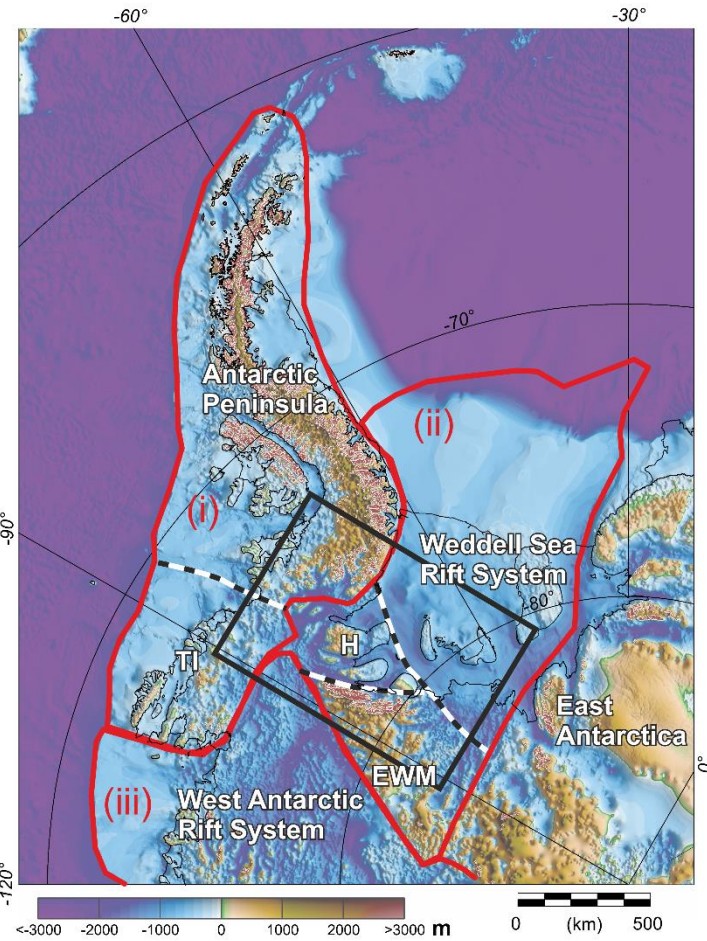

**Figure 3**: Subglacial topography from BedMachine (Morlighem et al., 2020) overlain with key West Antarctic geological provinces (red outlines) including (i) Antarctic Peninsula-Thurston Island (TI), (ii) Weddell Sea Province, and (iii) West Antarctic Rift System-Marie Byrd Land (Jordan et al., 2020). Dashed black lines mark sub-province boundaries, including the Haag block (H) and Ellsworth Whitmore Mountains (EWM). Black box locates Figure 1.

The majority of the West Antarctic crust developed between 500 and 90 Ma, along an active ocean-continent convergence zone that bordered the ancient Pacific margin of Gondwana (Jordan et al., 2020). During the break-up of Gondwana, the four major crustal units making up the exposed rocks of West Antarctica, namely the Antarctic Peninsula, Thurston Island, the Ellsworth-Whitmore Mountains (EWM) including the Haag Block and Marie Byrd Land (Figure 3), moved outwards from the margin of the East Antarctic craton towards the convergent Pacific margin (Dalziel and Lawver, 2001). This movement of crustal blocks during the early stages of Gondwana breakup may have been driven by an upwelling mantle plume, triggering



a major pulse of magmatism and the break-up of the Gondwana supercontinent (Jordan et al., 2017; Jordan et al., 2020; Maslanyj and Storey, 1990).

The EWM block is a large crustal block allochthonous with respect to other parts of West Antarctica (Dalziel et al., 1987; Maslanyj and Storey, 1990), the origin of which is subject to some debate. Some interpretations suggest that the EWM block
was originally located in a position ~1,500 km further north adjacent to both the Antarctic and South African margins, with rotation into its current position taking place during Late Triassic or Early Jurassic crustal shearing as Gondwana break-up initiated (Curtis and Storey, 1996; Dalziel, 2013). Others propose a less far travelled model, where the EWM block was located 500 km northeast of its current location and was rifted from East Antarctica by extension and transferred to West Antarctica (Jordan et al., 2017).
Regardless of the uncertainties around the initial position of the EWM block, its movement resulted in an extensional rift system (the Weddell Sea Rift System) that developed between the moving Haag-Ellsworth-Whitmore block and East Antarctica. This extensional rifting created major fault systems, with extended and modified crust forming the floors of the Weddell Embayment (Dalziel and Lawver, 2001). The thermal age of the stretched continental crust below the Filchner-Ronne ice shelf has been dated to 230 to 165 Ma, which is in good agreement with the timing of Gondwana break-up (Jones et al.,
2002; Studinger and Miller, 1999). Therefore, the broad assembly of the microcontinental blocks (Haag, EWM, Antarctic Peninsula) in the Evans-Rutford region (Figure 3) was likely to have been completed within the Jurassic, with the latest time being approximately 165 Ma, the youngest thermal age of the crust underlying the Filchner-Ronne Ice Shelf.

Following this Jurassic assembly, thermochronometry data indicates that the Ellsworth Mountains experienced significant exhumation during the Early Cretaceous between ca. 145 Ma and 117 Ma, uplifting by 4 km or more (Fitzgerald and Stump,
1991, 1992), following the initial separation of East and West Gondwana, closely following the opening of the Weddell Sea. Subsequent post-Early Cretaceous uplift and denudation has been constrained to a maximum of 3 km. However, it has not been distinguished whether this occurred gradually or in a short-lived pulse (Fitzgerald and Stump, 1992, 1991). Following assembly, the landscape has been modified by subaerial, and subsequently subglacial, processes.

**1.2 Geomorphological evidence of erosional surfaces beneath the Antarctic Ice Sheet**

An understanding of the past landscape evolution of a region can be gained by investigating remnants of ancient landscapes preserved beneath the ice sheet. The preservation of pre-glacial landscape remnants beneath the Antarctic Ice Sheet is most likely where the direction of ice flow under a selective erosion regime remains consistent over multiple glacial cycles. Under such conditions, warm-based erosive ice remains focussed in pre-existing valleys, and thin, slow-flowing and cold-based ice can be present on adjacent areas of higher topography (Sugden, 1978). Therefore, ancient landscapes can be preserved under
cold-based ice throughout the Cenozoic despite having been subject to, for example, 58 glacial cycles during the last 13 Ma (Naish et al., 2009).

Features associated with selective linear erosion are flat surfaces with intervening troughs, which suggest the inheritance of pre-glacial topographic structure and provide insight into the co-evolution of the regional landscape and ice sheet (Sugden,



1978). Low-relief flat surfaces reflect potential relict non-glacial surfaces that have been preserved in the landscape
(Goodfellow, 2006), in contrast to the overdeepened subglacial troughs which reflect the location of past erosion (Cook and
Swift, 2012). Moreover, the presence of linear landscape breaks, steep-sided basins, and sharply-bounded planar surfaces
(tilted or otherwise) can be used to infer the presence of likely geological faults (Burbank and Anderson, 2012).

## 2 Approach

To understand the landscape evolution of the Evans-Rutford region, the objective of this study was to conduct
geomorphological analysis of the landscapes recorded in the RES data. We used the picked radar line data in preference to the
BedMachine DEM (Morlighem et al., 2020) for three reasons that are specific to our study and type of analysis. Firstly, our
focus in this study is mapping the detailed extent and morphology of geomorphological features with prominent edges such as
troughs and plateaux. Close inspection of the difference between BedMachine and our picked line data show that the line data
better represents the edges of features, which are often smoothed in BedMachine (Figure 4).
Secondly, our approach requires hypsometric analysis, and so necessitated the minimisation of interpolated elevations that
might smooth any frequency modes or 'peaks' in the hypsometry. Thirdly, part of the area of our study is covered by the
TORUS survey which does not appear to be included in the BedMachine compilation (Morlighem et al., 2020). All of our
quantitative analysis is carried out only on the picked and interpolated line data (see Section 2.2) but we use BedMachine in
figures to illustrate broad topographic patterns. The subglacial topography derived from the interpolated RES surveys was
analysed to classify the landscape of the study region (Jamieson et al., 2005), identify the geomorphic processes that have
operated in the region, and reconstruct how the present subglacial topography formed.



**Figure 4**. Difference between BedMachine bed elevation and our picked bed elevation radar line data from GRADES, TORUS and Evans surveys. Legend shows different interpolation methods used in compiling the BedMachine topography (Morlighem et al., 2020). Black line illustrates the grounding line derived from BedMachine (Morlighem et al., 2020). Colour ramp shows the difference between BedMachine and our picked line data. Reds show areas where Bedmachine topography is lower, blues where BedMachine is higher. Inset shows detail of mid-Rutford and Carlson Inlet. The biggest differences occur around the edges of plateaus and troughs and in the area of the TORUS survey. Inspection of individual radargrams shows that prominent breaks in topography are smoothed in BedMachine and so we use our line data in our analysis.



## 2.1 RES datasets

The Twin Otter aircraft used during the GRADES-IMAGE survey was equipped with the British Antarctic Survey's PASIN (Polarimetric radar Airborne Science INstrument) radar system (Corr et al., 2007; Frémand et al., 2023). The aircraft was
equipped with dual-frequency carrier-phase GPS for navigation (with absolute GPS positional accuracy of 0.1 m), wing-tip magnetometers, and a radar altimeter for surface mapping. The instrumentation and initial processing of the RES data are outlined in Corr et al. (2007), Jeofry et al. (2018) and Frémand et al. (2023), with key aspects summarised below.

The PASIN system was operated with a centre frequency of 150 MHz and used two interleaved modes: a 4 μs, 10 MHz bandwidth linear deep-sounding chirp, and a 0.1 μs unmodulated shallow-sounding pulse. Along-track data were collected at
a post-processing trace spacing of 20 m and vertical resolution of 8.4 m (Frémand et al., 2023). Compression of the chirp data, best used to assess the bed in deep ice conditions, was applied using a Blackman window to minimise sidelobe levels, and then processed using a coherent averaging filter along a moving window of length 15 (Corr, 2021). SAR focusing was not applied to the data. Surface elevation was derived from the radar altimeter for ground clearance less than 750 m, and from the PASIN system for higher altitudes. Bed elevation values presented here are all referenced to the WGS84 ellipsoid.

The bed reflector was semi-automatically picked from the chirp data, using a first break picker below a top-mute window of ~100 samples above the approximate bed reflection in the PROMAX software package (Frémand et al., 2023). In the event of multiple reflections at the ice-bed interface, the first reflection was assumed to be the bed except in areas where earlier reflections clearly represented areas of entrained debris or obvious processing artefacts (Frémand et al., 2023). Whilst the pick data were manually checked, the semi-automatic bed pick algorithm applied to the GRADES-IMAGE survey did result in
some gaps and errors in the data which were corrected where identified. However, due to the size of the dataset (27,550 line-km) not all pick errors will have been identified. We manually reviewed and corrected the published GRADES-IAGE data for erroneous bed reflection picks using the software ReflexW, which applied to approximately 10% of total line length. Most commonly, this applied to sections where the semi-automatic picker produced a bed pick but where no clear bed reflection was apparent in the RES data. Sources of error within the picked bed reflections can arise from surface clutter returns masking
the echoes of the ice-bedrock transition at depth, or diffraction hyperbolae caused by an unfocused return of the energy. These would make the ice-bed interface reflection unclear and difficult to identify (Frémand et al., 2023; Napoleoni et al., 2020).

Picked travel time was converted to depth using a radar wave velocity through ice of 168 m/μs (as used in all BAS data; Frémand et al., 2023). A constant firn correction of 10 m was applied to correct for uncertainties in the speed of electromagnetic waves through the firn layer (e.g. Ross et al., 2012).

To increase the spatial coverage and point density of the bed elevation picks within the region of study, bed-pick data from the BAS 1994/95 Evans and 2001/02 TORUS (Targeting ice stream Onset Regions and Under-ice Systems) RES surveys were also used (Corr, 2020; Corr and Smith, 2020; Frémand et al., 2023). Combined, these two surveys collected a total of ~20,000 line-km of RES data over Evans and Rutford ice streams using the coherent "BAS-built" radar system, a low-power



predecessor of the PASIN system that operated with the same frequency and pulse modes as PASIN (Corr and Popple, 1994;
Frémand et al., 2023).

Exposed bedrock datapoints were also added from the Reference Elevation Model of Antarctica (REMA) (Howat et al., 2019),
so that exposed sub-aerial local and regional-scale bedrock features such as nunataks or mountain ranges were included in the
interpolated data points. These data were extracted from REMA v1 (Howat et al., 2019) using a bedrock mask derived from
automated analysis of Landsat8 imagery (Burton-Johnson et al., 2016). The combined RES and subaerial datasets created a
bed elevation point dataset covering ~50,000 km of line data spanning large parts of the southern Antarctic Peninsula and
WAIS (Figure 1).

## 2.2 Interpolation of RES lines

Interpolation of bed elevation datasets derived from RES surveys have utilised a range of gridding algorithms, for example,
natural neighbour (Ross et al., 2012; Young et al., 2011), kriging (Aitken et al., 2014; Rippin et al., 2004, Fretwell et al., 2013;
Jeofry et al., 2018), and minimum curvature gridding (Bianchi et al., 2003). Additionally, gridding techniques using tension
spline algorithms have produced realistic DEMs of similar subglacial topography close to our study region (Paxman et al.,
2018, 2019; Napoleoni et al., 2020). Such splining methods are commonly used with irregularly-distributed data confined to
discrete transects in order to create a grid of the inferred subglacial bed topography (Young et al., 2011). Given the irregular
distribution of our survey data, we applied a continuous bivariate cubic spline algorithm to interpolate the radar lines. This was
carried out in order to better illustrate the more precise datapoints given from the direct observations of the bed. The
interpolated radar lines were generated at a spatial resolution of 500 m, and because of higher uncertainty with increased
distance away from the directly surveyed points, we masked the interpolation at a radius of 2 km from each bed elevation point
location in a similar method to Napoleoni et al. (2020) and Paxman et al. (2019). Whilst not producing a continuous DEM, the
interpolation of the radar lines provides a more direct measurement of the morphology of the bed topography.

## 2.3 Terrain analysis

In order to understand landscape evolution, we used our buffered interpolated data to map macro-scale landforms across the
study area, with additional interpretation of radargrams to assess detailed topographic morphology along profiles. This initial
classification was aimed at identifying the presence of flat surfaces, troughs and alpine glacial features within the study area.
We delineated valleys and troughs as linear features with high relief (Jamieson et al., 2014). Flat surfaces were mapped as
continuous low-angle surfaces with low local relief which are characterised by a sharp break in slope at their edges with lower
elevation valleys on at least one steep side (Wilson and Luyendyk, 2006).

Troughs surrounding the surfaces were separated into different morphological types. Firstly, the widest troughs with more
gently sloping trough walls were identified as U-shaped from the RES cross-profiles. Potential fault-bounded troughs were
identified where linear, steeply-sloping trough walls with excavation at the base were evident. Troughs with two thalwegs
close to the slope breaks and a central topographic rise, giving the floor a W-shaped profile, were also identified. V-shaped





troughs - often asymmetric - were commonly situated between surfaces, with intersecting RES profile lines showing V-shaped cross-sections with steep slopes and clear thalwegs. The thalwegs were mapped using profile lines across each channel to find the deepest section.

Hypsometric analysis was also undertaken to analyse the frequency-distribution of elevations and highlight where there are

significant areas of land at particular elevations. The distribution of area versus altitude can offer opportunities to study the dominance of particular processes within a landscape (Jamieson et al., 2014, Rose et al., 2013). For example, the presence of U-shaped valleys cut to a particular elevation, or the presence of flat surfaces across a region might result in a peak in area-elevation distribution because of the relatively flat valley flows of those glacial troughs. Hypsometry was therefore calculated for the interpolated radar lines.

**2.4 Analysis of pre-glacial topography**

Given the possibility that remnants of pre-glacial topography can survive within the modern subglacial landscape (Sugden and Jamieson, 2018), we generated an ice sheet-unloaded version of the topography in order to identify landscape components that may have existed prior to glacial inception. To do so, we used a model that accounts for the isostatic response to the complete unloading of the Antarctic Ice Sheet (Paxman et al., 2022). We added this correction to the present-day bed, yielding rebounded

topography under ice-free conditions, which was then subjected to hypsometric analysis. The isostatic adjustment model does not account for vertical movements arising from the isostatic response to glacial erosion or glacio-tectonic interactions.

## 3. Results

### 3.1 Flat surfaces

We identified 10 distinct flat surfaces across the Evans-Rutford region between the Antarctic Peninsula and the Ellsworth

Mountains (Figure 5). The area of each surface ranges from 650 to 11,032 km$^2$ (Table 1). Mean flat surface elevations are referenced to the WGS84 ellipsoid, and range between -476 m and 165 m. Clusters of similar mean elevations across multiple surfaces are also evident. For example, surfaces 7 and 9 have mean elevations at -62 m and -60 m respectively (and are located close to one another in the south-western part of the study area). The lower-lying surfaces 4 and 8 have mean elevations of -476 m and -456 m, respectively. Surfaces 1, 2 and 3, which are close neighbours in the northern part of the study area, sit at

mean elevations within a range of less than 50 m. Mean slopes across the surfaces are all <5 °. The mean flow speed of the ice overlying the surfaces ranges between 7.3 m yr$^{-1}$ and 22.8 m yr$^{-1}$, with mean ice thicknesses ranging between 654 m and 1252 m. Two potential flat surfaces beneath the Korff and Skytrain Ice Rises, at broadly similar elevations to the other identified surfaces, are also indicated by additional exploratory RES data.





**Figure 5:** Regional subglacial topography of the Evans-Rutford region. The background is BedMachine (Morlighem et al., 2020) but with our interpolated radar lines (outlined in black) of the Evans-Rutford region surveys overlain, along with mapped plateau surfaces and troughs.

| Plateau surface ID | Area (km²) | Mean elevation (m) | Standard deviation of elevation (m) | Mean slope (°) | Mean ice velocity from MEaSUREs (m yr⁻¹) | Mean ice thickness (m) | Mean rebounded elevation (m) |
|---|---|---|---|---|---|---|---|
| 1 | 650 | -237 | 157 | 3.0 | 11.5 | 1037 | 106 |





| 2 | 1333 | -197 | 325 | 4.0 | 14.4 | 849 | 65 |
|---|---|---|---|---|---|---|---|
| 3 | 1123 | -199 | 207 | 3.8 | 22.8 | 685 | -2 |
| 4 | 3249 | -476 | 179 | 1.0 | 10.3 | 1252 | -140 |
| 5 | 1832 | 126 | 108 | 1.8 | 7.3 | 654 | 313 |
| 6 | 11032 | 165 | 272 | 2.6 | 11.4 | 786 | 433 |
| 7 | 1619 | -62 | 148 | 1.5 | 10.5 | 918 | 267 |
| 8 | 5079 | -456 | 112 | 0.7 | 11.4 | 785 | -346 |
| 9 | 2562 | -60 | 336 | 4.6 | 12.3 | 596 | 112 |
| 10 | 7612 | -342 | 160 | 1.4 | 12.4 | 799 | -194 |

**Table 1:** Zonal statistics calculated for each individual plateau surface derived from interpolated radar lines. Overlying ice flow velocity derived from MEaSUREs (Rignot et al., 2017).

The RES data suggest that the majority of the ten 'plateau' surfaces are extremely flat (Figure 6a, b). In contrast, the topography of surface 6 is more variable, with a small peak on its eastern side and low relief peaks and troughs across its surface. The
margins of surface 6 are also more crenelated in comparison to the linear margins visible in the other surfaces (Figure 5). The subglacial topography of the Fletcher Promontory consists of two surfaces (9 and 10) situated at two different elevations (200 m and -400 m respectively) separated by a step with a relief of ~600 m (Figure 6c, d).





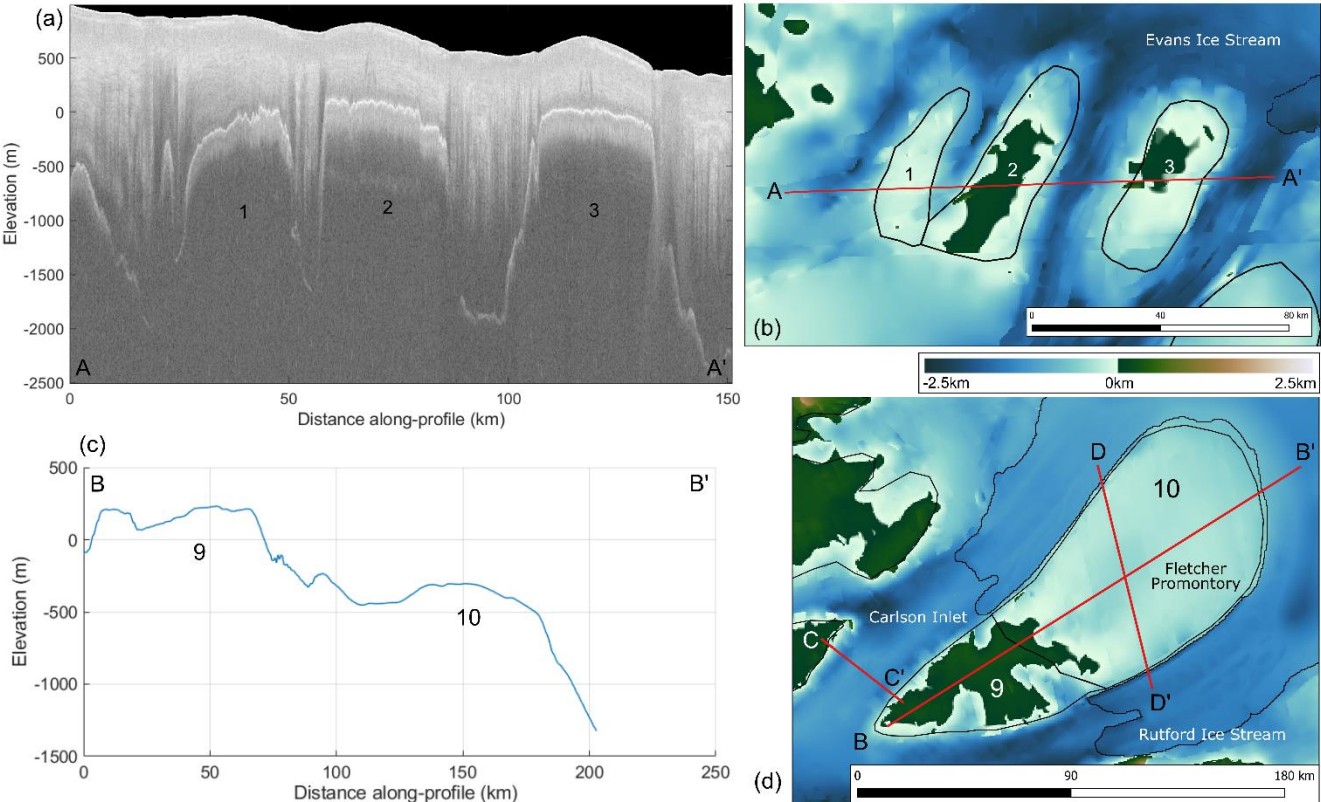

**Figure 6:** (a) Radargram illustrating plateau surfaces 1-3 along profile A-A'. (b) Profile line (A-A', displayed in red) over
plateau surfaces 1-3. (c) Interpolated elevations from BedMachine subglacial topography data (Morlighem et al., 2020) along
profile B-B'. BedMachine data were used for this profile as there were no directly overflown flightlines from the GRADES-
IMAGE RES survey. (d) Profile line (B-B', displayed in red) over the Fletcher Promontory, plateau surfaces 9 and 10. Profile
lines C-C' and D-D' illustrate the locations of radargrams in Figures 7 and 9.

## 3.2 Troughs

Throughout the Evans-Rutford region three distinct trough morphologies are present. Troughs in the study area are linear
valley features, mostly V- or W-shaped with abrupt, steep margins, and some gently sloping U-shaped. Some of these troughs
are presently occupied by fast-flowing ice, reaching a maximum of 641 m yr$^{-1}$ in the Evans Ice Stream, whereas some contain
slow-flowing ice (e.g. ~0.2 m yr$^{-1}$ in Carlson Inlet). The Rutford Ice Stream and Carlson Inlet, which surround the Fletcher
Promontory (Figure 5), are hosted by troughs that appear to be fault-bounded. The cross-profiles of these troughs have a W
shape (Figure 7), as the subglacial bed is deepest directly adjacent to the base of the trough walls with a raised ridge in the
centre of the trough. The trough walls are linear and reach a maximum slope of ~46°.





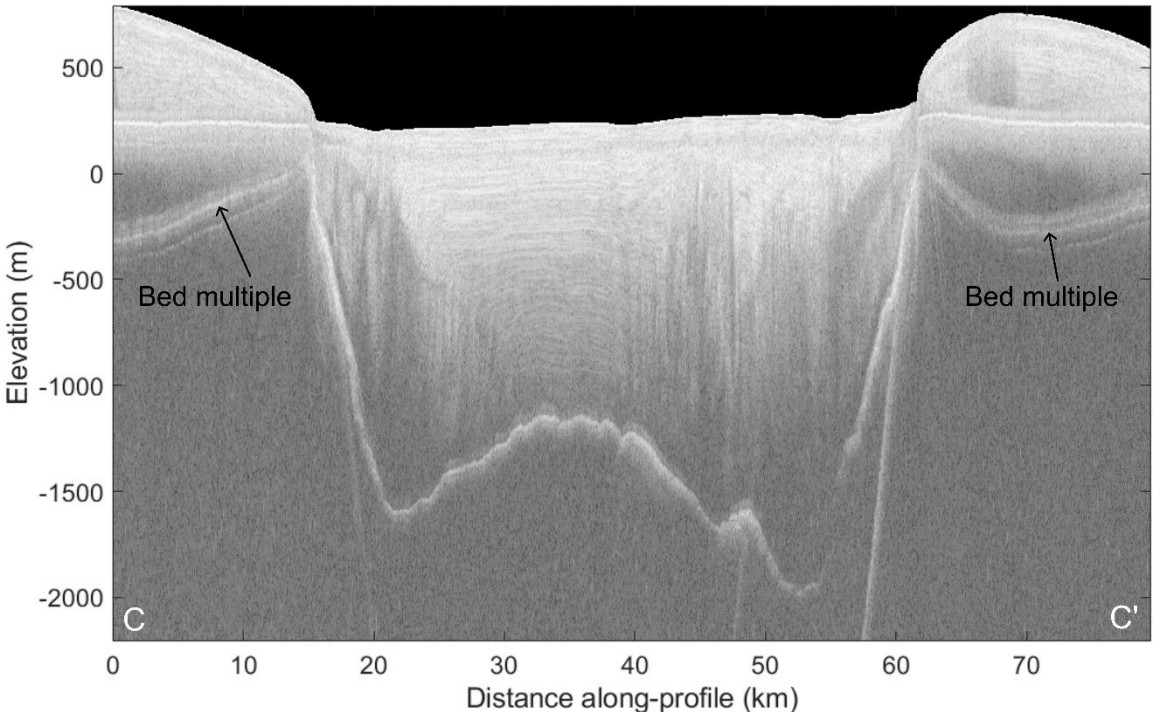

**Figure 7:** Radargram demonstrating the W-shaped cross-profile in the Carlson Inlet trough. Location of C-C' profile line is illustrated in Figure 6d. Bed multiples are due to an additional reflection between initially transmitted radar wave with ice surface and the aircraft.

V-shaped asymmetric troughs are present between surfaces (e.g. Figure 6a), with one containing the Talutis Inlet (Figure 1). The thalwegs of these troughs are offset from the centre of the valley, with a steeper trough wall closer to the line of the thalweg. The widths of these troughs average between 10.2 km to 12.1 km across the top, with maximum relief ranging from 1398 m to 2424 m. Wide U-shaped troughs contain the tributaries of the Evans ice stream, ranging between 20 km to 70 km at their widest point.

**3.3 Hypsometry**

The hypsometric curve derived from the interpolated radar lines contains three significant peaks in hypsometry at 171, -328 and -728 metres (Figure 8b). The non-rebounded dataset (Figure 8a) illustrates where the areas contributing to these hypsometric peaks occur within the landscape. The highest elevation surface population is situated between -50 m and 500 m, whilst the mid-elevation surface population ranges from -550 m to -50 m. The lowest elevation hypsometric peak is associated with the geographic east side of Evans Ice Stream, and the onset zone of Rutford Ice Stream. The elevations below all three





hypsometric peaks can be attributed to the deeper western inland side of Evans and its tributaries, as well as the troughs containing Rutford Ice Stream, Carlson Inlet, and Talutis Inlet.

In contrast to the three peaks in the present-day subglacial topography, the hypsometry of the rebounded topography (Figure 8c) only demonstrates two significant peaks. The higher of these lies between 182 m and 582 m, with a modal value of 382 m. The lower peak in hypsometry is most prominent at -217 m, but ranges between -417 m and -17 m. This indicates that some of the previously separate populations in the non-rebounded topography merge into a smaller number of populations when the landscape is rebounded. The majority of the flat surfaces have remained in the same population as in the non-rebounded

hypsometry, although the hypsometric peak containing the lower surfaces has merged with the peaks representing the lower elevations of the troughs.







**Figure 8**: (a) Interpolated radar lines displayed in colour-categorised symbology relating to hypsometric peaks. Plateau surfaces are outlined in black, and these and the line data are displayed over hillshaded BedMachine subglacial topography

map (Morlighem et al., 2020). (b) Hypsometric curve of the line data classified according to significant peaks (coloured bars on right elevation scale correspond to colours in panel (a)). Values of significant peaks are marked. (c) Hypsometric curve derived from the isostatically-rebounded Digital Elevation Model of line data with significant peak values marked.

## 4 Discussion

### 4.1 Extensive (and ancient?) palaeo-surfaces beneath the Evans-Rutford Region

Our mapping and hypsometric analysis shows for the first time that a number of regionally coherent low-relief surfaces exist beneath the WAIS in the Evans-Rutford region. These surfaces have remarkably constant elevation and gentle slopes with minimal relief. The hypsometric peaks of the surface populations occupy a narrow range of elevations, and show modal values of 171 m for the higher elevation surfaces (Surfaces 1, 2, 3, 5, 6, 7, 9) and -328 m for the lower surfaces (Surfaces 4, 8, 10). When rebounded, the elevations of these hypsometric peaks shift upwards to 382 m and -217 m respectively. Given that there

is no obvious way to form such regional surfaces under conditions of glacial erosion, the similarities between surfaces allow us to propose that they represent remnants of pre-glacial surfaces that were originally laterally continuous over 100s of kilometres. The preservation of such topography is plausible via the mechanisms of selective linear erosion in the troughs that bound the surfaces, allowing regional ice to discharge and cold-based preservation to occur across these surfaces (Sugden, 1968). The internal structure of the layering visible in the RES data (Figure 9) appears flat and continuous/undisturbed

throughout the ice column overlying the surfaces, which lends further support to the idea of non-erosive ice conditions. The lateral continuity of the englacial layering suggests that this low velocity pattern has been stable through time, at least for as long as the current ice column has been accumulating (Siegert et al., 2003; Matsuoka et al., 2015).



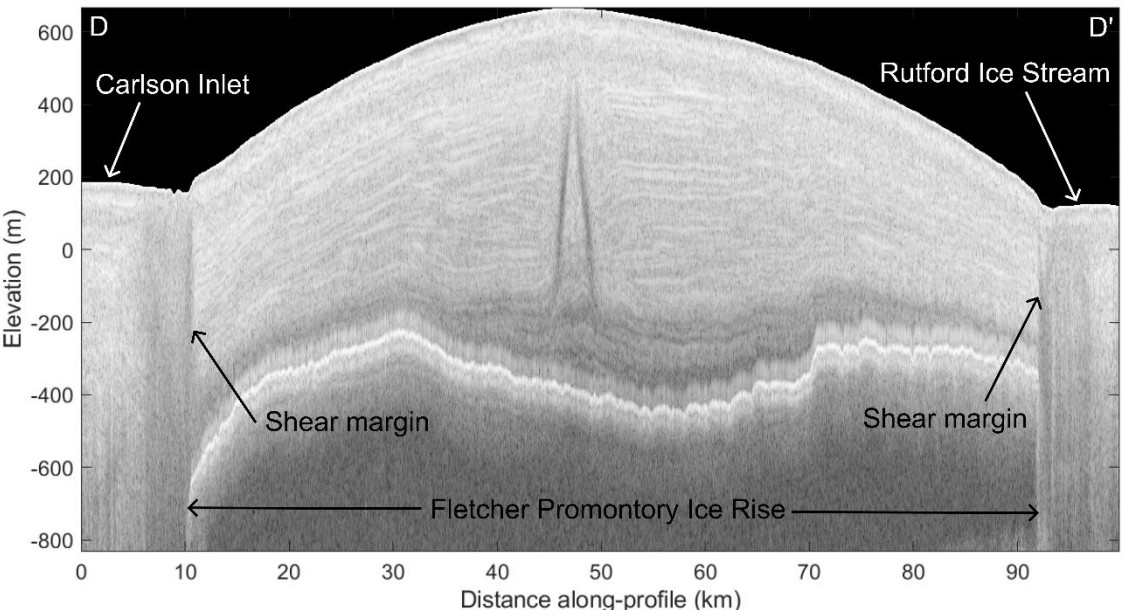

**Figure 9:** Radargram illustrating continuous undisturbed layering overlying the Fletcher Promontory flat surface. Location of
D-D' profile line is illustrated in Figure 6d. The centre of Fletcher Promontory demonstrates a well-developed Raymond Bump,
indicating an ice-divide triple junction (Hindmarsh et al., 2011).

The presence of two populations of surfaces is suggestive of downwearing of the upper surface to form a second, lower,
surface. The morphology of the Fletcher Promontory (surfaces 9-10) is an example of this, where there is a 600 m change in
elevation at an obvious 'step' between surfaces 9 and 10 (Figure 6c). A similar pattern is evident further east between surfaces
6 and 8. The more seaward surfaces, (8 and 10), make up the lower surface (illustrated in white in Figure 8a) and are also at a
similar elevation to the surface under Korff Ice Rise, which sits within the same hypsometric peak. The higher elevation
surfaces (1, 2, 3, 5, 6, 7 and 9; Figure 8a) sit landward of the lower surface.

Similar extensive flat surfaces have been identified using RES data elsewhere in Antarctica including in the Wilkes Subglacial
Basin (Paxman et al., 2018), the Weddell Sea Embayment (Rose et al., 2015), the Ross Embayment (Wilson and Luyendyk,
2006), and Marie Byrd Land (LeMasurier and Landis, 1996). Common features of these surfaces are that they are laterally
continuous over 10-100 km scales and demonstrate a flat and level (or at least low-angle) nature with low relief, often separated
by linear troughs. Interpretations of the formational processes of these surfaces have involved marine erosion or wave action
into the basement in the absence of glacial ice (Wilson and Luyendyk, 2006), or passive margin evolution with a retreating
escarpment dividing two surfaces following the Gondwana break-up (Paxman et al., 2018). Prolonged fluvial erosion down to
a peneplain has also been proposed, the processes of which may have interplayed with marine planation (LeMasurier and
Landis, 1996; Rose et al., 2015). A model of high-elevation, low-relief landscape formation as a result of river network





disruption (Yang et al., 2015) could potentially apply, however, this seems unlikely given how extensive these surfaces are across the region. In any case, the erosional surfaces represent prolonged intervals of erosional levelling that must have taken

place in a relatively stable tectonic environment (Beaumont et al., 2000, Brown et al., 2000).

## 4.2 Regional extent of flat surfaces

The Evans-Rutford flat surfaces are not the only ones to have been identified beneath the WAIS. The identification of similar pre-glacial surfaces preserved beneath upstream the upstream regions of the neighbouring Institute and Möller ice streams (Rose et al., 2015), approximately 400 km to the south of our study area, indicate that these surfaces may extend across

microplate boundaries, implying a timing of formation that was after plate convergence and assembly in the Jurassic-Cretaceous. The geometry of the Evans-Rutford region is comparable to the gently sloping, low relief topographic block of the Institute-Möller region, which comprises two surfaces separated by a distinct break in slope. Marine and fluvial peneplain erosion models have been presented for the formation of these surfaces (Rose et al., 2015).

As is the case for our flat surfaces, the Institute and Möller erosional surfaces comprise a lower seaward surface and a higher

landward surface. The Institute–Möller surfaces exhibit hypsometric peaks at 900 m and 400 m below sea level, a vertical offset of ~500 m (Rose et al., 2015). These similarities are consistent with a similar formational process acting more widely around the Weddell Sea Embayment.

## 4.3 Geological structures in the Evans-Rutford region

The morphology of the troughs containing Rutford Ice Stream and Carlson Inlet demonstrate a W-shaped cross-profile, with

near-vertical trough walls and a topographic ridge running centrally down-trough, aligned with the trough axis (Figure 7). The troughs are also very linear and have abrupt changes in slope where they meet the flat surfaces. The features of these troughs can be characterised as representing a fault-bounded trough, similar to those found in the George VI Sound (Smith et al., 2007). These linear bedrock depressions bounded by steep escarpments are indicative of faults (Garrett et al., 1987), where glacial erosion has been focused along fault-damaged rock at the base of the trough wall. Similar deep subglacial troughs have been

described on either side of the Shackleton Range in East Antarctica, where the fast-flowing glaciers reflect the trend of, and have exploited, the subglacial fault systems (Paxman et al., 2017, Sugden et al., 2014).

Primarily, the troughs have been identified as a structural feature or tectonic valley, the topography of which has been subsequently modified by glacial activity (Smith et al., 2007). The coincidence of this morphology in Rutford and Carlson troughs suggests a common process. The nature of the fault-bounded flat surfaces may suggest a block-faulted landscape,

which can be produced by extension in rifted basins (Burbank and Andersen, 2012).

## 4.4 At what elevation might these surfaces have formed?

The isostatically-rebounded interpolated radar lines reinforces the interpretation of two pre-glacial laterally-continuous surfaces (Figure 8c). The fact that isostatic rebound reduces the number of hypsometric peaks would be most consistent with





two plateau surfaces forming pre-glacially, but differential ice loading subsequently offsetting some of the surfaces. However,
the isostatic adjustment model may not take into account all other processes that may have subsequently affected the elevation
of these surfaces.

Processes capable of vertical displacement of the surfaces include processes driving uplift and some driving subsidence. Where
selective erosion occurred within the glacial troughs, and with relatively little erosion occurring on the flat surfaces themselves,
the flexural isostatic response to erosional unloading would have uplifted the flat surfaces (Summerfield, 1991). The pattern
of this effect would be dependent upon the amount of glacial erosion and the flexural rigidity of the lithosphere. This region
of West Antarctica is characterised by relatively low flexural rigidity, resulting in shorter wavelength variability in isostatic
uplift (Paxman et al., 2022, Swain and Kirby, 2021). Additional flexural uplift may have also been driven by mechanical
unloading of the footwall by the hangingwall in any troughs bounded by extensional faults, as demonstrated in the Shackleton
Range region (Paxman et al., 2017), although the sense of motion on the faults in the Evans-Rutford region is unclear. Both of
these processes would have acted to increase the elevations of the flat surfaces. In other words, the pre-glaciation elevation of
the flat surfaces was likely lower than the ice-rebounded hypsometric peak values of 382 and -217 m, meaning that the higher
elevation population of surfaces may have resided closer to palaeo-sea level at the inception of the Antarctic Ice Sheet. Our
rebounded topography does not incorporate tectonics or the effects of rebound due to erosion, and so we are unable to quantify
these effects. However, the surfaces in this region do not show systematic tilting, as is observed in the Shackleton Range,
suggesting that fault-driven upwarping may be less important.

Conversely, two processes that may have caused post-formation *subsidence* of the surfaces include long-term thermal
subsidence post-dating West Antarctic and/or Weddell Sea rifting (Wilson and Luyendyk, 2006), and long-term sediment
loading in the Weddell Sea to the north and east (Hochmuth et al., 2020). A third process which is not well constrained, but
can cause long-wavelength uplift or subsidence, is the effect of dynamic topography (Austermann et al., 2015). Therefore, it
is likely that the pre-glacial elevations were different to the simple ice-unloaded values. The potential effects of these uplift
and subsidence processes in the landscape evolution of the region are discussed further below.

## 5. Possible models of Evans-Rutford landscape evolution

The relative age of some of the events that formed the Evans-Rutford landscape can be discerned by the relationships of the
different landscape elements. Firstly, the faults must pre-date the flat surfaces as the surfaces are not displaced across the faults.
Secondly, the lower surface post-dates the higher surface into which it is eroded. Thirdly, the glacial troughs must post-date
the surfaces since they separate surfaces of accordant elevation. Given this sequence of events and what is known about the
geological evolution of the region, as described in Section 1.1, potential hypotheses for the formation of the flat surfaces are
presented below.





### 5.1 Passive margin evolution

The first potential mechanism for the formation of the flat surfaces is through passive margin evolution (Burbank and Andersen, 2012), where the upper surface forms through the initial topographic uplift and erosion associated with the continental break-up of Gondwana, with subsequent post-rifting scarp retreat forming the lower surface. The most prominent landforms resulting from continental break-up are high elevation passive margin escarpments. For example, the central Namibian margin demonstrates similar topographic elements of a well-defined major escarpment with a relief of 1000 m,

separating a gently inclined coastal plain from an interior plateau (Cockburn et al., 2000). Passive margin retreat on a similar lateral extent over 100s of kilometres, as well as a similar timescale of movement post-Gondwana break-up, can be seen in the continental rifting between South America and Africa, which began during the late Jurassic (Brown et al., 2000). The lateral extent of the flat surfaces in the Evans-Rutford region compares closely to the scale of the features in Namibia, with the escarpment in Namibia retreating around 160 km (Cockburn et al., 2000), whilst the lower surface of the Fletcher Promontory

spans approx. 130 km. An Antarctic analogy can also be drawn to the McMurdo Dry Valleys mountain block, which contains two escarpments with relief over 1000 m. As a result of initial lithospheric extension and rifting, fluvial denudation to a new sea level has removed ~4 km of material from the coast in that area, accompanied by isostatic uplift and faulting (Sugden and Jamieson, 2018). This has left a coastal plain 100 km wide, backed on the inland margin by an escarpment.

Marine planation forming erosional surfaces has been suggested in the Ross Embayment, creating plateaux and terraces sitting

350 m to 100 m below sea level, separated by bedrock troughs occupied by West Antarctic ice streams (Wilson and Luyendyk, 2006). In contrast, one of the models proposed for bedrock surfaces in the Institute and Möller catchments of the Weddell Sea involves fluvial peneplain erosion, during or after the rifting and break-up of Gondwana around ~180 Ma, although another proposed mechanism is also marine planation (Rose et al., 2015). In response to the downward shift in base level caused by the rifting of Gondwana (Beaumont et al., 2000), fluvial systems incised into the landscape, forming broad, extensive coastal

surfaces backed by erosional escarpments.

Similar formational processes could have created the geomorphology seen in the Evans-Rutford region, as there is a lower, seaward surface, backed by a landward higher elevation surface, with a relief of at least 600 m between them (Figure 6, 8).

However, the lower surface sits below present-day sea level (and also therefore below higher palaeo-sea level in an ice-free world) when isostatically rebounded for ice sheet removal (-217 m), therefore requiring other processes to have since led to

subsidence. Potential mechanisms for this include: 1) prolonged regional thermal subsidence that will have occurred after uplift resulting from the main magmatic event in the region at ~180 Ma (Sleep, 1971), or 2) sediment loading in the Weddell Sea, where sediment deposition in offshore basins that evolved after the break-up of Gondwana would likely have led to local subsidence (Hochmuth et al., 2020). Dynamic topography is another process by which subsidence may have taken place, as evolving viscous stress and buoyancy associated with mantle convection flow can drive vertical deflections of the crust

(Austermann et al., 2017, 2015) but is currently poorly constrained for Antarctica.





**5.2 Planation and uplift associated with the West Antarctic Rift System or Antarctic Peninsula**

The other hypothesis for flat surface formation is through the planation of an extensive surface throughout the region by marine erosional processes, similar to those proposed for the Norwegian strandflat surface (Fossen, 2023). The surface would then have been uplifted in association with the West Antarctic Rift System (WARS) or cessation of subduction along the margin of

the Antarctic Peninsula, allowing subsequent incision of a second lower elevation surface. These processes would have to have occurred after the Jurassic assembly of the microplates (Jordan et al., 2020) and exhumation of the Ellsworth Mountains (Fitzgerald and Stump, 1991), with sea level remaining stable in order to allow for the prolonged planation that would result in an erosional surface of this nature.

Similar low-relief erosion surfaces occur in West Antarctica and New Zealand, representing prolonged intervals of erosional

levelling in a stable tectonic environment (LeMasurier and Landis, 1996). These surfaces are compatible with prolonged fluvial erosion, marine planation, or a combination of both, with the New Zealand surface being found to consist of a composite of features. The data from these surfaces does not support a continuous low-relief landscape pre-breakup (LeMasurier and Landis, 1996), however, they can provide a clear comparison. Erosion surfaces in western Marie Byrd Land have also been identified, which formed around ca. 70-60 Ma corresponding with slow cooling following activity in the West Antarctic Rift System

(Zundel et al., 2019).

Rock uplift in the EWM block in the Early Cretaceous may have been caused by early rifting in the WARS between the EWM and Marie Byrd Land (Fitzgerald, 2002). The WARS saw further activity in the Cenozoic and ongoing fracturing, volcanic activity, and bordering uplift associated with development of the rift combined to modify the topography of West Antarctica (Dalziel and Lawver, 2001). This may have created enough uplift in the Evans-Rutford region to initiate erosion of the lower

surface via either marine or fluvial erosion.

Alternatively, the collision of spreading ridge segments at the continental ocean boundary of the Antarctic Peninsula triggered the progressive northward shutdown of subduction along the Antarctic Peninsula margin after 90 Ma, which could have resulted in uplift in the Evans-Rutford region (Jordan et al., 2020, Larter and Barker, 1991). The initial updoming associated with the West Antarctic and New Zealand surfaces at 85-75 Ma may have then been succeeded by uplift further north along

the Antarctic Peninsula as subduction ceased between 83 and 45 Ma.

Long term subsidence following rifting, or other potential mechanisms that would lead to subsidence as described in Section 5.1, would consequently lead to the lower elevation surfaces now residing below sea level, as indicated by the flexural isostatic rebounding model.

Differentiating between an origin of passive margin evolution or of WARS associated planation and uplift is challenging.

However, the presence of erosional surfaces in the Institute and Möller catchment on the eastern side of the Evans-Rutford region give an indication that the surfaces were formed through passive margin evolution, as the other hypotheses involve processes of uplift occurring on the western margin, distally to the Rose et al. (2015) surfaces. Given the coincidence of these surfaces and other similar flat surfaces across Antarctica, for example in Marie Byrd Land (Wilson and Luyendyk, 2006), the





Wilkes Basin (Paxman et al., 2018), and West Antarctica and New Zealand (LeMasurier and Landis, 1996), we propose that
passive margin evolution associated with fragmentation of the Gondwanan supercontinent may be a fundamental process
driving formation of these flat surfaces. We note that similar sets of surfaces are found around other continents formerly part
of Gondwana (Ollier, 2014).

## 5.3 Controls on long-term ice sheet flow

Erosion of the structurally-controlled troughs lying between the flat surfaces is likely to have occurred after their planation,
forming conduits for the ice streams observed today. After onset of the WAIS, incision would have followed the structural
weaknesses in the bedrock, which happen to be largely aligned with radial ice flow from the central WAIS (Bingham et al.,
2012; Jamieson et al., 2014). Therefore, these faults would act as conduits for ice during the initiation of Antarctic glaciation,
providing topographic steering (Bingham et al., 2012; Kessler, 2008) and promoting fast, warm-based ice flow that would
have strongly influenced the long-term structure and evolution of the ice sheet (Aitken et al., 2014).

Assuming the flat surfaces are pre-glacial, the flat surfaces themselves are therefore indicators of a long-term relatively stable
thermal and ice flow regime over those areas of WAIS. To have survived, the surfaces must have been preserved beneath cold-
based, slow flowing ice. In periods of expanded ice (for example, the LGM) these surfaces could not have been significantly
modified. Under periods of potentially reduced WAIS extent, such as the Pliocene, it is plausible that some of these flat surfaces
could act as independent ice dispersal centres and, if they rebounded above sea level enough, as potential stabilization points
for advancing ice shelves and grounded ice during periods of ice re-advance (Bradley et al., 2015; Matsuoka et al., 2015) but
without significant erosion.

## 6 Conclusions

In this study we facilitated the first in-depth investigation of the large-scale subglacial topography of the Evans-Rutford
Region. We combined bed picks from the newly released BAS GRADES-IMAGE RES survey, alongside the TORUS and
Evans 1994/95 RES surveys. From geomorphological analysis of these RES surveys, we conclude:

- Two populations of flat surfaces identified across the region suggest two extensive regionally coherent surfaces,
  which can be interpreted as pre-glacial landscape remnants lying at similar elevations. The topography of the region
  suggests the incision of the lower surface into the higher surface, given the two populations lying at different
elevations and the stepped topography evident on the Fletcher Promontory.

- The surfaces are likely to predate inception of the WAIS, and the step between them likely formed as part of passive
  margin evolution following Gondwana break-up.



- Deep troughs have been incised between surfaces through selective linear erosion during the glaciation of the WAIS,
  potentially due to the exploitation of structural weaknesses in the bedrock that have resulted from faulting. This has
  controlled the long-term pattern of ice discharge from the WAIS.

- The preservation of these pre-glacial flat surfaces is an indication of where cold-based, non-erosive ice has prevailed
  for much of the WAIS history. The flat surfaces also have the potential to impact patterns of ice sheet retreat and re-
  advance given their elevation and extent.

Sampling of the subglacial bedrock, along with potentially determining exhumation histories from thermochronometry, would
be required to better constrain the potential formational mechanisms of the flat surfaces within the Evans-Rutford Region.
Higher resolution gravity, magnetic or seismic surveys could also be used to identify variations in bedrock geology or constrain
geophysical modelling. Specifically, modelling of the gravity anomalies over the plateaus and incised troughs could give an
idea of the nature of their isostatic support, and the extent of uplift driven by glacial incision. Higher resolution magnetic data
would help constrain the location of faults and other major tectonic boundaries which we hypothesise as pre-conditioned areas
for more rapid erosion.

**Author contribution**

CMC, MJB and SSRJ conceptualised the study, with CMC carrying out the investigation under supervision from MJB and
SSRJ. GP, JB, NR, TJ and FN contributed to the development of the methodology and data analysis alongside CMC.
CMC prepared the manuscript with contributions from all co-authors.

**Competing interests**

The authors declare they have no conflict of interest.

**Acknowledgements**

This work contributed to NERC Grant NE/R002029/1. We are grateful to the scientists, pilots and support staff who collected
the radar data utilised here, especially the late Richard Hindmarsh who conceptualised the GRADES-IMAGE survey and to
Rob Bingham who designed the survey lines. MB's involvement was supported by funding received from the European
Research Council (ERC) under the European Union's Horizon 2020 research and innovation programme (Grant agreement
No. 885205).



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
