# Peer review of "Extensive palaeo-surfaces beneath the Evans-Rutford region of the West Antarctic Ice Sheet control modern and past ice flow."

_EGUsphere, 2023_

## Author Comment (AC1)

We thank the reviewers for their constructive comments on this manuscript. In the response below, we address the comments made by Referee #1 (David Sugden) and explain the changes we have made to the manuscript. Reviewer comments are in *black italics*, and our responses are in red.

RC1 (David Sugden)

*This is an excellent paper adding new insights into the bed of a significant part of the West Antarctic Ice Sheet. The paper describes, analyses and interprets the bed topography extending across parts of the southern Antarctic Peninsula, the West Antarctic Rift System and the Weddell Sea Rift System. Better knowledge of the bed of Antarctica is important both for understanding current ice sheet dynamics and in elaborating geological evolution. The scientific background to this is covered effectively.*

*What is exciting is that the authors discover two extensive and coherent flattish surfaces that are incised by steep glacial troughs. They argue that the surfaces are relict and have been protected beneath cold-based, slow-moving ice and that glacial erosion has been selective and largely confined to the troughs. The subglacial landscape predates ice-sheet glaciation in Antarctica and is reminiscent of similar landscapes studied in the formerly glaciated areas of the Northern Hemisphere although on a grander scale. The authors put forward two interpretations. First, the surfaces represent fluvial landscape evolution following the break-up of Gondwana, similar to that for example in Southern Africa. In such a passive margin environment, the high surface is the former interior surface of Gondwana and the lower surface is related to fluvial erosion to the new lower sea level following break up. The second hypothesis for flat surface formation is through the planation of an extensive surface throughout the region by marine erosion. They favour the first explanation.*

*The paper brings together new results from three extensive radio-echo sounding surveys and analyses the topography along the radar lines in order to pick up sharp contrasts in slope, such as an escarpment or steep trough wall. This approach is highly effective in identifying coherent landforms. Overall, the paper relies on quantitative analysis and the hypsometry results are particularly striking, especially when applied to an isostatically rebounded topography. The figures on both methods and the results are excellent and allow a reader to follow the approach clearly and in detail.*

*An additional point of interest is to find that a similar pattern of two surfaces of similar elevations have been observed 400 km away across a micro plate boundary on the flanks of Institute and Möller ice streams. The similarity supports the passive margin fluvial origin for the surfaces.*

We thank the reviewer for their kind and positive comments.

*I wondered whether they might comment on another way of establishing fluvial activity by the recognition of river valley patterns. I was struck by the dendritic pattern at the head of Evans Glacier (Fig 2c) and the angles of the confluences that are characteristic of fluvial activity. This pattern also features on either side of surfaces 7 in Fig 5. Also suggestive is the sinuous talweg of the valley between surfaces 3 and 4, a typical fluvial signal. Perhaps the crenulated margins of surface 6 is best explained by fluvial activity? Are any of these points worth a mention?*

At the beginning of Section 5.3., we have added some discussion on the likely inherited fluvial network. This introduction now reads:

"The current ice flow patterns are controlled by the inherited drainage patterns imposed by Gondwanan tectonic processes and by the original fluvial network that would have formed prior to Antarctic glaciation. We note the distinctive dendritic pattern and confluence angle of the valleys, particularly in the upper reaches of the Evans Ice Stream. The ice flow would likely have inherited the fluvial network and exploited it via selective linear erosion (Sugden, 1968, Sugden et al., 2014) leading to trough formation and flat surface preservation. Erosion of the structurally- and fluvially- controlled troughs lying between the flat surfaces is likely to have occurred after surface planation, forming conduits for the ice streams observed today. After onset of the WAIS, incision would have followed the structural weaknesses in the bedrock and pre-existing fluvial network, which both happen to be largely aligned with radial ice flow from the central WAIS (Bingham et al., 2012; Jamieson et al., 2014). Therefore, these troughs would act as conduits for ice during the initiation of Antarctic glaciation, providing topographic steering (Bingham et al., 2012; Kessler, 2008) and promoting fast, warm-based ice flow that would have strongly influenced the long-term structure and evolution of the ice sheet (Aitken et al., 2014)."

*The Discussion is full and careful in considering the two hypotheses.*

*As a geomorphologist, I find the marine hypothesis difficult to believe. Inherited from the mid-20th Century, there has been a view that marine erosion can erode extensive surfaces. But the problem is that the erosion is attributed to wave action and that on a slope from the coast to the sea, wave action is unlikely to be able to erode a platform more than a few 100s of metres across. Here we are talking of gently sloping surfaces measuring tens by hundreds of km. Marine planation as an extensive process of erosion would have to rely on an unlikely relationship with relative sea level over millions of years. Perhaps it would be clearer to argue that the change of relative sea level may allow rivers to erode a landscape to near sea level. Indeed, following plate tectonic separation, there are new coastlines and a passive margin situation is perfect for low-relief plains to form inland of and parallel to the new coast, as in Namibia which they quote. Having said this, I am all for the authors keeping discussion of both hypotheses in the paper – if perhaps nuanced.*

We have inserted and edited sentences at the end of the first paragraph in Section 5.2., which reads:

"We regard marine planation at this scale as unlikely, as there is no modern analogue for how surfaces of such breadth could be produced by wave action. For marine planation to have formed these surfaces as an extensive process of erosion, it would require an unlikely relationship with sea level over millions of years, in order for surfaces of this scale to have been formed.

Similar, but not as extensive, low-relief erosion surfaces occur in West Antarctica and New Zealand, representing prolonged intervals of erosional levelling in a stable tectonic environment (LeMasurier and Landis, 1996)."

*More details.*

*Line 134: I do like the way you deal with the difference between Bedmachine interpolation and your radar line approach.*

We thank the reviewer for their positive comment.

*Line 228: Floors not flows?*

Thank you for catching this; "flows" has been amended to "floors" here.

*Fig 8: Really interesting to see the effect of isostatic rebound on the hypsometry.*

We thank the reviewer for their positive comment.

*Line 338: You seem to push the wave cut hypotheses over large areas.*

Here we have rephrased the sentence at line 338, which now reads:

"Interpretations of the formational processes of these surfaces have involved passive margin evolution with a retreating escarpment dividing two surfaces following the Gondwana break-up (Paxman et al., 2018), or marine erosion or wave action into the basement in the absence of glacial ice (Wilson and Luyendyk, 2006)."

*Lines 416-418: The coastal plain here today is nearer 10 km wide rather than 100km. I would be tempted to add the reference to Fitzgerald, 1992 instead of Sugden and Jamieson, since this is the main source of the figure of 4 km, and cut the last sentence. The reference is: Fitzgerald, P.G. 1992. The Transantarctic Mountains of southern Victoria Land: the application of fission track analysis to a rift shoulder uplift. Tectonics,11, 634-662.*

We agree with this; the reference has been amended and the last sentence has been removed.

*Line 472: After Ollier. Add Summerfield, M.A., (Ed), 2000. Geomorphology and global tectonics. Wiley, 367 pp. for a comprehensive review?*

The reference suggested by the reviewer has been added here.

---

## Author Comment (AC2)

We thank the reviewers for their constructive comments on this manuscript. In the response below, we address the comments made by Referee #2 and explain the changes we have made to the manuscript. Reviewer comments are in *black italics*, and our responses are in red.

RC2 (Anonymous Referee #2)

*This paper discusses the formation process of subglacial topography using existing data from Bedmachine and acquired RES data. It suggests that the formation period of flat surfaces was before the formation of West Antarctic Ice Sheet, and suggests that the direction of ice flow is influenced by pre-glacial topography. It is an interesting study that provides insights into the process of ice sheet formation and the formation of subglacial landscape. It fits within the scope of the journal and is judged as publication.*

We thank the reviewer for their positive comments.

*There is a need to describe deeper into the correction for rebound. I think that the GIA model adopts a 2D structure of the Earth, but is it correct to understand that the parameters follow Paxman et al. (2021)?*

To describe the rebound correction in more detail, we have amended the sentence at Line 233: "To do so, we used a model that accounts for the isostatic response to the complete unloading of the Antarctic Ice Sheet (Paxman et al., 2022)", which now reads:
"To do so, we adopted a recent calculation of the isostatic response to the complete unloading of the Antarctic Ice Sheet, computed using a flexed elastic plate model with a laterally variable effective elastic thickness of the lithosphere (Paxman et al., 2022a). The effective elastic thickness in this model exhibits relatively low values of 10-20km in the Evans-Rutford region (Swain and Kirby, 2021)."

*In this case, the rebound values should vary by the model adopted, but how much variance is there? I would like that variance to be documented in Table 1.*
*Additionally, could the choice of Earth's structure significantly influence the results of the isostatic rebound elevation distribution and impact the discussion?*

We have added a paragraph at the end of the Results section (3.3. Hypsometry) in order to clarify this:
"The uncertainty in the rebounded elevations of the surfaces ranges from 42 m to 100 m (Paxman et al., 2022b). However, given that effective elastic thickness varies over relatively long wavelengths (> 100 km) in this region (Swain and Kirby, 2021), the surfaces will be near-equally affected by any uncertainty. Therefore, the uncertainty could have a minor effect (10s of m) on the elevation of the hypsometric peaks, but would not diffuse the peaks. The low effective elastic thicknesses used in the isostatic rebound calculation for the Evans-Rutford region (Paxman et al., 2022b) have been reported in multiple studies (Jordan et al., 2010, Chen et al., 2018, Swain and Kirby, 2021), and are at the lower end of the range used by Paxman et al. (2022b) to generate uncertainties. Therefore, we consider these uncertainties to be maximum values for this region."

*Specific and technical comments*

*Please cite the software used for creating the maps.*

> Maps were created using QGIS and the authors' own material, in some cases specifically Quantarctica, which has been cited in the figure caption of Figure 1. It is not journal policy to cite software used when the material used has been created solely by the authors.

*Line 118: Please remove the comma between subglacial and processes.*

> The comma between subglacial and processes has been removed here.

*Specify Marine Byrd Land in Figure 3.*

> We have realised that the figure caption here has an error, as Marie Byrd Land is for the majority not visible in the figure. Therefore, we have amended "West Antarctic Rift System-Marie Byrd Land" to "West Antarctic Rift System" in the figure caption.

*Line 186: An explanation of the abbreviation of TORUS has already appeared.*

> Thank you for spotting this; we have removed "(Targeting ice stream Onset Regions and Under ice Systems)" from the sentence.

*Adding a map to Figure 5 that makes it easier to identify flat surfaces (by narrowing down the elevations for coloring and enlarging the region) would make it clearer for the readers.*

> We have edited Figure 5 in order to highlight the flat surfaces: the colour scale is now limited to -1500 m to 1500 m, which highlights the elevations of the surfaces in relation to the surrounding subglacial topography. The sentence "The colour scale saturates at -1.5 km and 1.5 km, in order to highlight the elevations of the surfaces in relation to the surrounding topography" has been added to the figure caption. The new figure would look like this:

[Figure]

*In Figure 6, please arrange the map first and the cross-section afterwards, e.g., moving position A to B, and B to A.*

The positioning of the panels in Figure 6 has now been changed (panel a swapped with panel b, and panel c swapped with panel d). The figure caption and corresponding references to the figure in the text have also been changed, now reading: "**Figure 6:** (a) Profile line (A-A', displayed in red) over plateau surfaces 1-3. (b) Radargram illustrating plateau surfaces 1-3 along profile A-A'. (c) Profile line (B-B', displayed in red) over the Fletcher Promontory, plateau surfaces 9 and 10. Profile lines C-C' and D-D' illustrate the locations of radargrams in Figures 7 and 9. (d) Interpolated elevations from BedMachine subglacial topography data (Morlighem et al., 2020) along profile B-B'. BedMachine data were used for this profile as there were no directly overflown flightlines from the GRADES-IMAGE RES survey." The new figure would look like this:

[Figure]

*For Figure 8c, indicate which flat surfaces correspond to which peaks.*

Figure 8c now includes the numbers of the surfaces included within the hypsometric peaks. The new figure would look like this:

[Figure]

*In lines L282-L286, including V-shaped valley and Talutis Inlet on the map would make it easier for readers to follow.*

These labels have been added to Figure 6, see above response for visualisation of the new figure.